# Scalable Fabrication of Thermally Conductive Layered Nacre-like Self-Assembled 3D BN-Based PVA Aerogel Framework Nanocomposites

**DOI:** 10.3390/polym14163316

**Published:** 2022-08-15

**Authors:** Mohammad Owais, Aleksei Shiverskii, Artem Sulimov, Dmitriy Ostrizhiniy, Yuri Popov, Biltu Mahato, Sergey G. Abaimov

**Affiliations:** 1Center for Design, Manufacturing and Materials, Skolkovo Institute of Science and Technology, 121205 Moscow, Russia; 2Center for Hydrocarbon Recovery, Skolkovo Institute of Science and Technology, 121205 Moscow, Russia

**Keywords:** thermal management applications, nanocomposite aerogels, thermal conductivity, boron nitride, polyvinyl alcohol

## Abstract

In this study, three-dimensional (3D) polyvinyl alcohol (PVA)/aligned boron nitride (BN) aerogel framework nanocomposites with high performance were fabricated by a facile strategy. The boron nitride powder was initially hydrolyzed and dispersed with a chemically crosslinked plasticizer, diethyl glycol (DEG), in the PVA polymer system. The boron nitride and DEG/PVA suspensions were then mixed well with different stoichiometric ratios to attain BN/PVA nanocomposites. Scanning electron microscopy revealed that BN platelets were well dispersed and successfully aligned/oriented in one direction in the PVA matrix by using a vacuum-assisted filtration technique. The formed BN/PVA aerogel cake composite showed excellent in-plane and out-of-plane thermal conductivities of 0.76 W/mK and 0.61 W/mK with a ratio of BN/PVA of (2:1) in comparison with 0.15 W/mK for the pure PVA matrix. These high thermal conductivities of BN aerogel could be attributed to the unidirectional orientation of boron nitride nanoplatelets with the post-two days vacuum drying of the specimens at elevated temperatures. This aerogel composite is unique of its kind and displayed such high thermal conductivity of the BN/PVA framework without impregnation by any external polymer. Moreover, the composites also presented good wettability results with water and displayed high electrical resistivity of ~10^14^ Ω cm. These nanocomposites thus, with such exceptional characteristics, have a wide range of potential uses in packaging and electronics for thermal management applications.

## 1. Introduction

The thermal management of devices is becoming increasingly critical as the demand for smaller electronic equipment and higher integration density grows. In today’s electronic sector, better manufacturing technology has led to miniaturization and integration, which has unavoidably increased the heat dissipation conundrums. The difficulty of thermal dissipation is exacerbated by the accumulation of heat, which reduces the longevity, dependability, and performance of electronics dramatically. As a result, the development of new materials with greater thermal conductivity (TC) has been prompted by the need to eliminate heat. Heat is mostly transported through phonons in the majority of polymers. However, the bulk polymer’s low TC (0.1–0.2 W/mK) restricts its use in applications that require quite high TC and rapid heat dissipation [1]. However, more study is being performed on continuous segregated thermally conductive filler networks since they are better at increasing TC than random filler distributions because of more effective phonon transfer channels [2,3,4]. Thus, the fabrication of continuous heat conduction pathways has been considered an efficient application of 3D-segregated structures in thermal interface materials with the fillers, distributed at the interfaces between polymer granules in a segregated design and with frequent micrometer-sized domains. These selective sizable amounts of filler exclusively at the polymer granule interface result in a continuous thermal network that begins to emerge at relatively low to optimum filler loading concentrations. Therefore, to receive an efficient contribution from these fillers, continuous and strong filler networks with a low specific surface area and interfacial thermal resistance are needed. Hence, it is obvious that polymers with better crystallite orientation and crystallinity, higher chain alignment, lower entanglement, and stronger inter-chain interactions have higher thermal conductivities [3,5].

Recent research has revealed that 2D materials with strong TCs, such as graphene [6,7], graphene oxide [8,9,10], MXene [11], boron nitride (BN) [12,13], and others, have tremendous potential in thermally conductive composites. Hexagonal boron nitride (h-BN) is a promising thermal conductive material with a graphene-like structure. It can easily exfoliate to generate boron nitride nanosheets (BNNS), which are two-dimensional (2D) nanosheet structures with good TC, great electrical insulation, and stability, meaning it is a promising ceramic filler.

The filler’s high TC is related to both its inherent TC and its thermal transfer efficiency per unit mass. The interfacial thermal resistance is one of the most important elements impacting thermal transfer efficiency; hence, thermal transfer efficiency per unit mass follows the trend of three dimensions > two dimensions > one dimension [14,15]. As a result, a three-dimensional (3D) interconnected network is advantageous for achieving a high thermal transfer efficiency per unit mass, which improves TC [16]. Putting BNNSs together in a 3D network structure is a good way to create a heat-conducting material with a high thermal transfer efficiency per unit mass. With a variety of matrix materials, certain research groups have proved that BNNSs create a 3D interconnected network. Zeng et al. suggested an efficient method for fabricating 3D BNNTs/rGO aerogels with good dispersion of BNNSs and improved TC, elasticity, and fatigue resistance [17]. Ding et al. employed the hydrogen bond between Boron Nitride (BN) and Polyvinyl alcohol (PVA) to control the creation of a 3D linked network in thermoplastic polyamide-imide materials to accomplish a sufficient BN dispersion and operate as a high thermally conductive network [18]. Teo et al. created OH-BNNS/PVA interpenetrating hydrogels by cyclically freezing/thawing an aqueous mixture of PVA and highly hydrophilic OH-BNNS nanocomposites with high TC, which have the potential to stabilize the BNNS dispersion and address mechanical failure and local overheating issues. The 3D structure is considered excellent for boosting heat management capability, according to the studies mentioned above [19]. Furthermore, given the present trend of highly integrated, lightweight, and portable electronic devices, using lightweight aerogels with 3D networks has a lot of potential [20].

Aerogel is a 3D network material that has ultralow density, high porosity, is lightweight, and has a changeable surface chemistry, meaning it is ideal for BNNSs to form the 3D skeleton achieving high thermal transfer efficiency per unit mass by tuning its porosity. A self-assembly method is frequently used to create aerogel [21], a light, porous network which has shown remarkable TC increase when used as a continuous network of fillers to create thermally conductive composites [22,23]. Briefly, 3D network aerogels are very appealing for a variety of applications, including environmental remediation, absorption, biomedical scaffolds, and supercapacitors, due to their combination of exceptional features. Incorporating BNNSs into the 3D structure of an aerogel can improve not only the heat transfer efficiency per unit mass of BNNSs but also the TC and mechanical properties of the aerogels [24,25,26,27]. The largest benefit of using BNNSs with aerogel comes from this multi-physical interdependence. Due to their high aspect ratio and low percolation threshold, BN platelets or sheets are favored among the thermally conductive fillers. The creation of this filler network and phonon routes is directly influenced by the ordering of such 2D fillers in a 3D system. It is preferable to align these fillers along the direction of heat flow in order to increase TC because they are anisotropic in nature, and the orientation or alignment of these 2D materials in a polymer network to form self-assembled 3D aerogels is a great way to improve the TC of the composite. For instance, Wu et al. aligned the BN flakes in BN/PVA-based composites, utilizing ice-templated self-assembly and hot-pressing techniques [28]. The formed aerogel obtained a high TC of 10.04 W/mK at a BN loading concentration of 68 wt.%. Typically, by these techniques, BN flakes are segregated and forced to line up at the ice interfaces by the directional growth of ice crystals [29]. Then, the BN-based PVA aerogels are squeezed into a circular plate using the hot press technique, helping to reduce porosity and align the BN flakes in one direction. Moreover, other approaches to BN alignment inside polymer, primarily in the horizontal direction, include the electrospinning technique [30,31], doctor blade casting [32], magnetic field alignment [33], injection molding [34], vacuum-assisted filtration technique, etc.

In our work, we fabricated facile and scalable 3D aerogel cake frameworks with highly aligned BN platelets in a PVA polymer system via the vacuum-assisted filtration technique, reaching excellent in- and out-of-plane TCs and maintaining electrical insulation of the polymer composites. The composite exhibited excellent thermal stability and wettability behavior, which is extremely pertinent for thermal management applications in the electronics industry.

## 2. Experimental

### 2.1. Materials

Boron Nitride (BN) powders with an average particle size of 10 µm, Polyvinyl alcohol (PVA) with an MW of 31,000–50,000, 98% hydrolyzed, and Diethyl glycol (DEG) were purchased from Sigma Aldrich Co., Ltd. (Burlington, MA, USA). All reagents were used without any further purification.

### 2.2. Characterizations

A scanning electron microscope (Thermo Scientific™ Helios G4 PFIB, Waltham, MA, USA) working at an accelerating high voltage allowed the observation of the microstructure of aerogels while an energy dispersive spectrometer (EDS) was used for the analyses of the elemental composition. The aerogels surfaces were carefully sputtered with gold particles prior to image observation. The water contact angle (WCA) was measured at room temperature using an optical contact angle analyzer to determine the wettability of the specimens. An adhesive tape was used to adhere the aerogels to a glass coverslip, and then 5 μL of deionized water was poured on the surface as an indicator for the WCA measurements. Each sample had at least three measurements, from which the average WCA values were determined. The functional groups on BN were identified by Fourier transform infrared (FTIR) spectrometry (Spectrum One, Perkin-Elmer, Waltham, MA, USA) in reflective mode. The observation was carried out within a wavenumber range of 4000–450 cm^−1^ for 10 scans per sample reading. The electrical volume resistivity of materials was tested using the Keithley 6517B and 8009 resistivity test fixtures (Tektronix, Beaverton, OR, USA). The in-plane and out-of-plane TCs of the samples were measured using a laser optical thermal scanner [35]. Testing was conducted on the cubic samples (surfaces polished) with the dimensions of (30 × 30 × 30) mm, and in this TC measuring technique, a flat surface of a specimen is heated by a focused, movable, and continuously operating optical heat source mounted with an array of three infrared temperature sensors, as described in [36]. The optical head, which houses the heat source and infrared sensors, moves at a constant speed relative to the sample, allowing the heater and sensors to follow scanning lines while keeping a constant spacing. The optical scanning method provides the TC measurements with both precision and accuracy of 2% (at a confidential level of 0.95) accounting for the heterogeneity and anisotropy of materials under studying. Thermal stability measurements of materials were studied by the Simultaneous Thermogravimetry-Differential Scanning Calorimetry analyzer STA 449 F3 Jupiter (Erich NETZSCH GmbH & Co. Holding KG, Selb, Germany) coupled with a Quadrupole Mass Spectrometer QMS 403 D Aëolos (Erich NETZSCH GmbH & Co. Holding KG, Selb, Germany). XRD measurements were observed on Aeris Benchtop X-Ray Diffractometer (Malvern Panalytical, Malvern, UK) with Malvern Panalytical Radiation of Cu K (alpha) 154,060 Å, step size 0.005° and range of 2θ from 5° to 85°, the XRD results were shown in Appendix A. A Perkin Elmer Lambda 1050 UV-vis-NIR spectrometer was employed to obtain the optical spectrum.

### 2.3. Methodology

A total of 15 wt.% PVA solution was prepared by PVA dissolved in distilled water at 105 °C for 1 h with continuous magnetic stirring. Next, it is mixed with 10 wt.% of Diethyl glycol (DEG) by stirring for 2 h at 105 °C. For the synthesis of BN suspensions, different BN weight percentages are dissolved in distilled water by mixing for 30 min via magnetic stirring followed by 3-h bath sonication. Consequently, the desired amount of BN suspension is added dropwise inside the existing PVA suspension. The suspension is then mixed for 1 h at 105 °C and poured on the vacuum-assisted filtration technique machine. After the filtration, the suspensions of PVA-BN aerogels are dried at room temperature for one hour, followed by overnight heating at 60 °C and by vacuum drying for 2 days. Samples of PVA-BN aerogels (Densities ~ 0.1 g/cm^3^) with various BN to PVA ratios (9:1, 9:3, 2:1) were fabricated via the vacuum-assisted filtration technique by following this procedure (Figure 1). The schematics of corresponding chemical bonding are presented in Figure 2.

## 3. Results and Discussion

The FTIR results (Figure 3) show two absorption peaks at 1366 cm^−1^ and 814 cm^−1^ due to the stretching and deformation of vibrations of B-N bonds, respectively, which are typically observed in the spectrum of BN. Moreover, a small peak of 2904 near 2926 may indicate the presence of impurities of C–H on the basal ends of BN. For BN-PVA aerogels, it was found that four bands at 3484, 2904, 1435, and 970 cm^−1^ represent the O–H stretching mode, C–H, C–O–H, and C–O bonds, respectively. The vibration of the O–H bond confirms the presence of surface hydroxyl groups on BN nanosheets by PVA. After the formation of BN-modified PVA aerogel, the O–H bond around 3484 cm^−1^ is observed with the band broadening of the C–O bond at 970 cm^−1^. These changes indicate the formation of hydrogen bonds between the hydroxyl groups of PVA and surface hydroxyl groups of BN nanosheets.

The elemental content of pure BN and BN/PVA composites showed a trend (Figure 4). The EDX spectra revealed that the peaks of C and O were slightly enhanced rather than remaining unchanged, indicating that the oxygenous and carbon groups were preserved. In comparison to pure fillers, the atomic composition of the BN/PVA composite reveals greater carbon and oxygen concentration. As demonstrated in Table 1 and Table 2, the O atomic ratio increased from 3.7 to 6.7 percent, while the C atomic ratio increased from 3.50 to 15.1 percent, indicating a strong chemical covalent bonding of PVA on the surface of pristine BN. With the robust aerogel being constructed via strong covalent bonding between BN, DEG, and PVA, the addition of oxygen and carbon-based functional groups attached to the BN surface is essential.

The wetting behaviors of materials are imperative to study for the applications related to thermal management, as thermal interface materials which are hydrophilic in nature may damage their effectiveness—water drops can be absorbed by pristine PVA aerogels due to their hydrophilic nature. BN with varied ratios of PVA was studied to modify the aerogels and improve their hydrophobicity in order to protect the porous structure of the aerogels and avoid moisture absorption. The rich hydroxyl groups of PVA aerogel (no BN) have a strong water adsorption capacity, leading to a water contact angle (WCA) of 0°. Varying the concentrations of BN in PVA as 9:3, 9:1, and 2:1, the WCA of aerogels increased to 135°, 119.2°, and 129.4°, respectively, indicating the hydrophobic surface of BN-modified PVA aerogels (Figure 5).

When compared to the pure BN-agglomerated cluster shown in Figure 6a, abundant BN nanoplatelets are adsorbed on the surface of each other via π-π- stacking, as demonstrated clearly in SEM images of Figure 6b–f. The improvement in the physical attributes of the BN/PVA-based polymer composite system, such as TC, is mostly due to the nanoplatelets’ alignment and stacking into BN clusters on top of each other inside the PVA polymer. This incredible alignment of BN platelets mimics the nacre-like structure, proving that it is the way for an effective route for the continuous passage of phonons, resulting in an increase in TC that was nearly isotropic in all directions. Due to the above facts, we believe that, although initiated by the vacuum-assisted filtration as a directed process, the alignment demonstrated some anisotropy, and its main part came from the platelet self-assembly, resulting in a near isotropic 3D skeleton.

The TC of PVA and BN/PVA-based aerogels with varied stoichiometric ratios is shown in Figure 7. The highest-reached TC of the BN/PVA-based aerogels is around 0.76 W/mK (in-plane) and 0.61 W/mK (out-of-plane) with the ratio of BN/PVA as (2:1), which is significantly higher than the TC of pure PVA 0.15 W/mK. Of course, the thermal conduction, which results from ordered heat-conducting channels, benefits from the highly aligned BN flakes in the PVA polymer system. The addition of BNNS to the PVA aerogel is known to strengthen the hydrogen and covalent bond interaction in the molecular structure, resulting in a considerable increase in TC. The PVA-DEG-BNNS cross-link interaction (as shown in Figure 2) would close the gap and connect the adjacent BNNSs, resulting in a considerable reduction in the interface thermal resistance and an increase in TC [37]. The morphology of BN platelets as shown in SEM images with a continuous alignment of BN in a PVA polymer system paved the way for an efficient transfer of phonons in a segregated BN filler-based PVA thermal channel. Other factors such as homogenous mixing, vacuum drying, and polishing the samples may have closed the gap between adjoined thermally conductive BN fillers in a closed segregated structure facilitating the reduction in interfacial thermal resistance. Correspondingly, BN/PVA-based aerogels with a stoichiometric ratio of (9:1) also exhibited an increase in the TC but did not achieve the same increase in TC in comparison to (2:1) in spite of higher BN content. Our idea behind investigating high BN:PVA ratios was to minimize the usage of low TC PVA, replacing it with a higher inherent TC of BN. This high filler content often leads to brittleness of the composite, and our aerogels (9:1) showed brittleness. The failure of the BN-PVA (9:1) aerogel to demonstrate a high TC 3D structure could be due to different factors. This decrease in TC might be the result of a nearly 90% increase in BN content in the PVA polymer, which causes the agglomeration and clustering of BN platelets. The agglomeration, in turn, causes an increase in interfacial thermal resistance, which lowers the TC and reduces phonon propagation. Hence, the enhancement in TC is limited due to the poor interfacial interaction between the filler and the polymer, the lack of polymer to create heat conductive bridges in-between BN particles and agglomerates, and, as the result, the construction of a poor thermally conductive adjacent network. On the other hand, in the intermediate case of a stoichiometric ratio BN-PVA (9:3), a considerable increase in TC was observed when the BN content in the PVA polymer reached ~ 66%. Due to the possible good uniform dispersion of BN inside PVA and homogenous mixing, BN-PVA showed good TCs in both planes. Hence, a strategy to obtain thermally conductive aerogel with an optimum stoichiometric ratio was introduced, where TC depends not only on the BN content but also on its structuring in the nanocomposite and on the possibility of phonon transfer from particle to particle by self-alignment and polymer bridges. Dropping the BN content to a ratio of (2:1), we did not observe a significant change in TC relative to (9:3), as shown in Figure 7, the fact demonstrating that high TC depends not as much on the BN content but on the optimal ratio and structuring on the nanolevel.

High electrical resistivity is an indispensable requirement for the composite materials used in thermal management applications employed for the electronic industry. Figure 8 depicts the electrical volume resistivity of neat PVA and its nanocomposites. As an electrical insulator, neat PVA has a high-volume resistivity of approximately 1.8 × 10^16^ Ω cm, but the addition of BN at various ratios reduces the volume resistivity to some minor extent. For example, with BN to PVA ratios of (9:1) and (9:3), the volume resistivity was found to be as low as ca. 1 × 10^14^ Ω cm and ca. 3 × 10^14^ Ω cm, respectively, whereas BN: PVA with a stoichiometric ratio of (2:1) nearly maintains the electrical insulation characteristics of neat PVA with an electrical resistivity of 1 × 10^15^ Ω cm. For all four cases, the electrical resistivity values lie well within the high-insulation range, indicating the straightforward applicability of the aerogels for TIM implementations in the electronics industry.

## 4. Conclusions

A facile technique was employed to construct a three-dimensional (3D) Polyvinyl alcohol (PVA)/aligned boron nitride (BN) aerogels nanocomposite framework with outstanding performance. As compared to the pure PVA matrix (thermal conductivity 0.15 W/mK), a highly aligned BN/PVA aerogel cake composite exhibited outstanding in-plane and out-of-plane thermal conductivities of 0.76 W/mK and 0.61 W/mK, respectively. Hence, we employed a strategy to acquire a thermally conductive aerogel with an optimum stoichiometric ratio, unaffected by the BN content, but with its structuring dependency on the nanocomposite framework with the possibility of efficient phonon transfer from particle to particle. In our study, the comparative analysis demonstrated that the highest TC was achieved not at the highest BN content, but in the range of BN-PVA stoichiometric ratios where the better self-structuring of filler and matrix material occurred. Furthermore, the composites demonstrated good hydrophobicity in water, high thermal stability, and a high electrical resistivity of at least 10^14^ Ω cm. As a result of their outstanding properties, these nanocomposites offer a wide range of potential applications in packaging and electronics for thermal management applications.

## Figures and Tables

**Figure 1 polymers-14-03316-f001:**
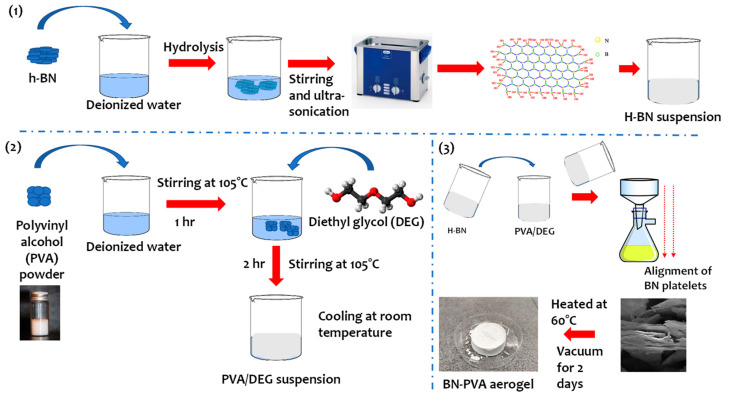
Schematics of the fabrication process of BN-PVA aerogels.

**Figure 2 polymers-14-03316-f002:**
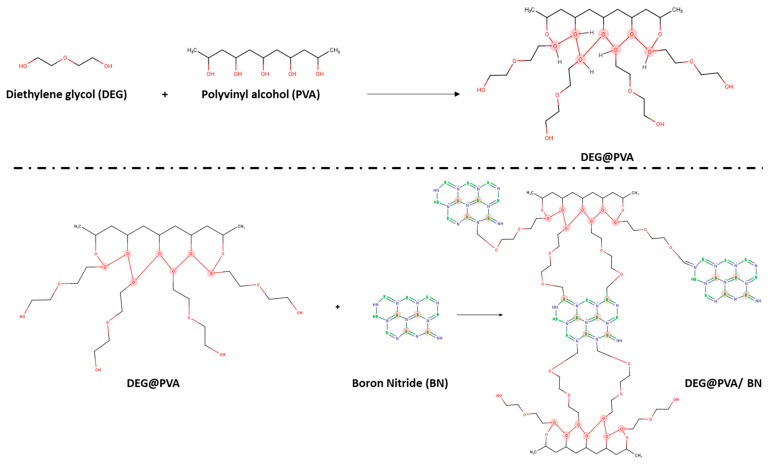
Schematics of the chemical bonding of DEG-modified PVA on BN in BN/PVA aerogels frameworks.

**Figure 3 polymers-14-03316-f003:**
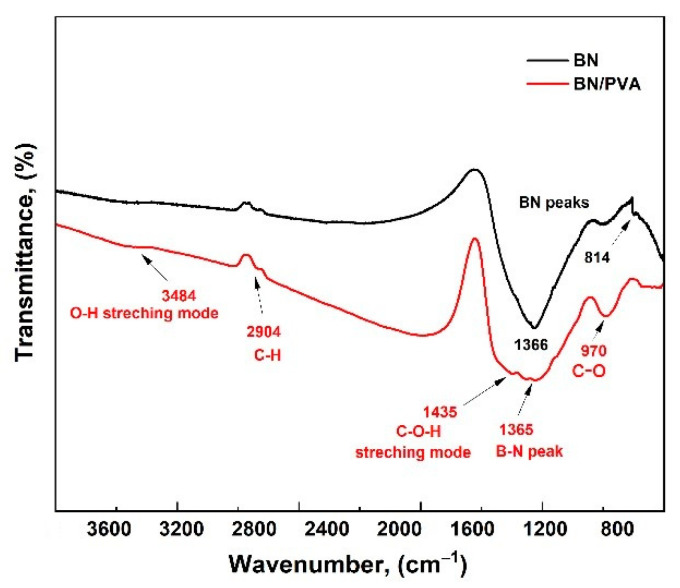
FTIR spectra of PVA aerogel and BN-modified PVA aerogel.

**Figure 4 polymers-14-03316-f004:**
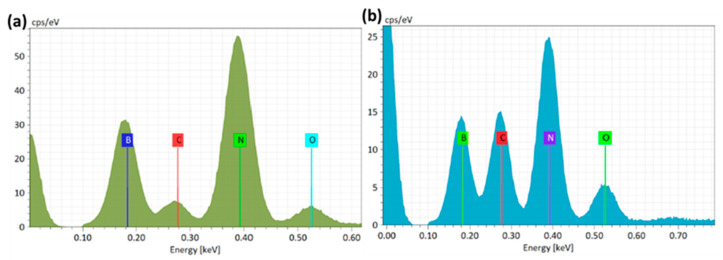
(**a**) EDX analysis of pristine h-BN composites, (**b**) EDX analysis of pristine h-BN/PVA composites.

**Figure 5 polymers-14-03316-f005:**
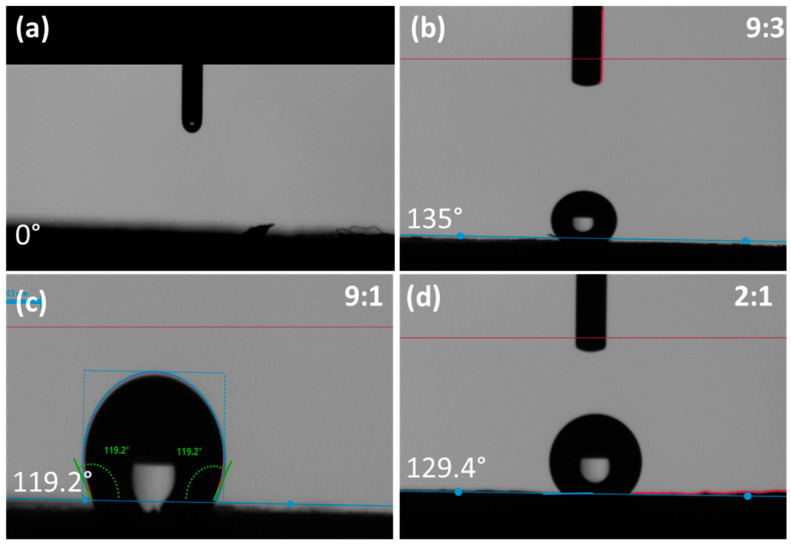
The WCAs of PVA aerogel and BN-modified PVA aerogels with different BN to PVA ratios; (**a**) 0:1; (**b**) 9:3; (**c**) 9:1; (**d**) 2:1.

**Figure 6 polymers-14-03316-f006:**
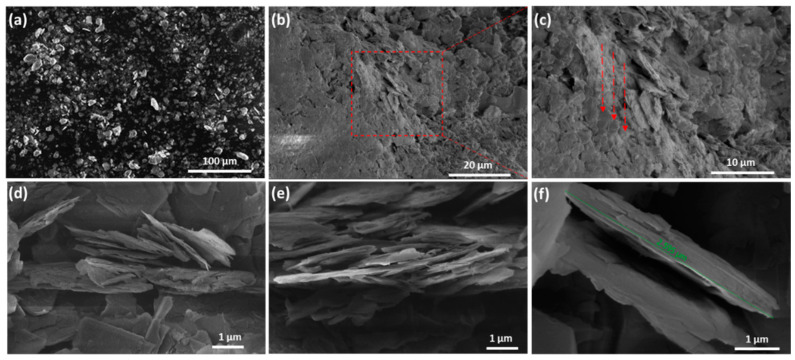
SEM of (**a**) pristine h-BN Powder; (**b**–**f**) Aligned h-BN/PVA composites at different magnifications.

**Figure 7 polymers-14-03316-f007:**
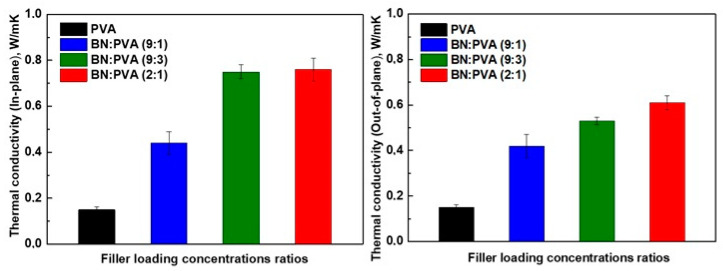
In and out of plane thermal conductivity of pristine PVA and h-BN/PVA composites.

**Figure 8 polymers-14-03316-f008:**
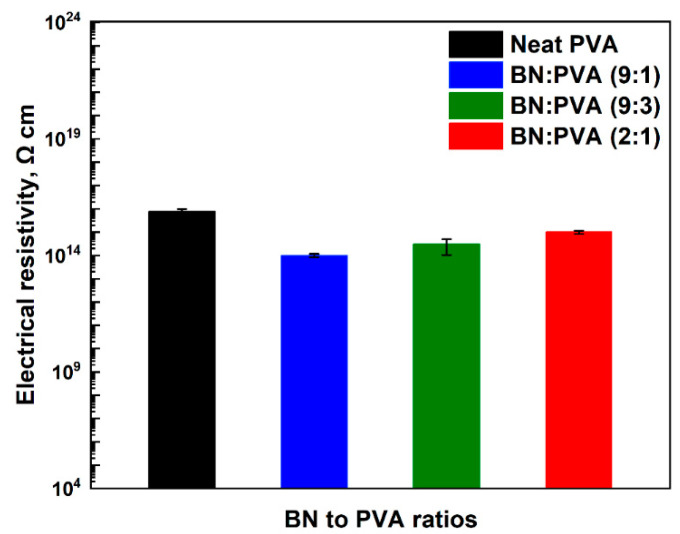
Electrical resistivities of pristine PVA and h-BN/PVA composites.

**Table 1 polymers-14-03316-t001:** Elemental composition of pure BN.

		Pristine BN			
Element	At. No	Mass (%)	Atom (%)	Abs. Error (%) (1 Sigma)	Rel. Error (%) (1 Sigma)
Nitrogen	7	53.92	48.47	5.77	10.70
Boron	5	38.10	44.38	4.19	11.00
Oxygen	8	4.63	3.65	0.59	12.73
Carbon	6	3.34	3.50	0.43	12.80

**Table 2 polymers-14-03316-t002:** Elemental composition of BN/PVA composite.

		BN/PVA Composite			
Element	At. No	Mass (%)	Atom (%)	Abs. Error (%) (1 Sigma)	Rel. Error (%) (1 Sigma)
Nitrogen	7	45.40	40.99	5.01	11.04
Boron	5	31.84	37.25	3.65	11.47
Oxygen	8	8.42	6.66	1.05	12.51
Carbon	6	14.34	15.10	1.65	11.51

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
