# Peer review of "Scalable Fabrication of Thermally Conductive Layered Nacre-like Self-Assembled 3D BN-Based PVA Aerogel Framework Nanocomposites"

_polymers, 2022, doi:10.3390/polym14163316_

Round 1

Reviewer 1 Report

In this manuscript the authors present their design and fabrication of electrically insulating yet thermally conductive polymer composites through 3D polyvinyl alcohol (PVA)/ aligned boron nitride (BN) aerogels via vacuum assisted filtration technique. The concluding results would provide good reference for further efforts in the area. The subject is worthy of interest and some of the presented solutions can indeed make a good contribution to this field. However, there are a few issues that should be addressed before it can be accepted for publication. Detailed comments are as follows:

Their nanocomposites demonstrated good hydrophobicity in water, high thermal stability, and a high electrical resistivity. This is somewhat not the case for many base materials. So more explanations are needed for readers to understand well the reason.

The present background introduction about the polyvinyl alcohol (PVA)/ aligned boron nitride (BN) composites is not sufficient or accurate. Quite a few closely related topical articles on polymer composite especially thermal interface materials were unfortunately missing. Some latest comprehensive review articles may help some such as: a) Advanced Energy Materials 11 (35), 2101387, 2021 and b) Polymers for Advanced Technologies 27 (11), 1484-1493, 2016 and more other approaches through adding other particles into liquid metal etc.

The authors emphasized the high TCs when they increase BN. Is there any unique virtue of this approach or its limitation?

Why 9:1 and 9:3 of BN:PVA ratio is the optimum? Please supply the relevant data

Author Response

Dear Reviewer,

Reviewer 2 Report

Comments and Suggestions for Authors

A three-dimensional (3-D) polyvinyl alcohol (PVA)/aligned boron nitride (BN) aerogels nanocomposite frameworks were prepared using a facile technique. The samples were tested for their structural, thermal and electrical characteristics using various techniques. The results showed outstanding in-plane and out-of-plane thermal conductivities, good hydrophobicity in water, high thermal stability, and a high electrical resistivity of at least 1014 â„¦ cm. Therefore, the reviewer considers that the manuscript is attractive to the readers in the research field of the polymer composites, and recommends the manuscript to be published in polymers after reconsidering following points.

1. Authors claimed the prepared PVA/BN aerogel nanocomposite is three-dimensional (3-D). Is there any evidence. Please explain it.  

2. Please provide more details of TC measurements in section 2.2. Characterizations.

3. What is the purpose of adding Diethyl glycol (DEG) in this work ?

4. Line 113, please make spaces µm and number.

5. Section 2.3, make spaces ℃& g/cm3 and number. Line 148, Remove dot mark in wt.%.

6. To better “indicate the formation of hydrogen bonds between the hydroxyl groups of PVA and surface hydroxyl groups of BN nanosheets”, the reference could be helpful:

Compos. Sci. Technol. 2021, 207.  DOI: 10.1016/j.compscitech.2021.108707

7. In figure 6, the picture information is incomplete, the author should check carefully.

8. Line 216, the sentence “Due to the last fact” should revised “Due to the above facts”

9. Please check the punctuation of the ordinate value in Figure 7.

10. Please check line 254, ca. 1×1014 â„¦ cm and ca. 3×1014Ω cm.

11. Please check the reference no.7 and no. 34.

12. “The formed BN/PVA aerogel cake composite showed an excellent in-plane and out-of-plane thermal conductivities…”Please mention the reason.

Author Response

Dear Reviewer,
